# Coupling Analysis of Compound Continuum Robots for Surgery: Another Line of Thought

**DOI:** 10.3390/s23146407

**Published:** 2023-07-14

**Authors:** Hangxing Wei, Gang Zhang, Shengsong Wang, Peng Zhang, Jing Su, Fuxin Du

**Affiliations:** 1School of Mechanical Engineering, Shandong University, Jinan 250061, China; hangxingwei@mail.sdu.edu.cn (H.W.); gangzhang@mail.sdu.edu.cn (G.Z.); pengzhang21@mail.sdu.edu.cn (P.Z.); jingsu@mail.sdu.edu.cn (J.S.); 2Key Laboratory of High-Efficiency and Clean Mechanical Manufacture of MOE, Shandong University, Jinan 250061, China; 3Shandong Center for Food and Drug Evaluation & Inspection, Jinan 250014, China; wangshengsong08@126.com

**Keywords:** coupling, compound continuum robot, inverse kinematics

## Abstract

The compound continuum robot employs both concentric tube components and cable-driven continuum components to achieve its complex motions. Nevertheless, the interaction between these components causes coupling, which inevitably leads to reduced accuracy. Consequently, researchers have been striving to mitigate and compensate for this coupling-induced error in order to enhance the overall performance of the robot. This paper leverages the coupling between the components of the compound continuum robot to accomplish specific surgical procedures. Specifically, the internal concentric tube component is utilized to induce motion in the cable-driven external component, which generates coupled motion under the constraints of the cable. This approach enables the realization of high-precision surgical operations. Specifically, a kinematic model for the proposed robot is established, and an inverse kinematic algorithm is developed. In this inverse kinematic algorithm, the solution of a highly nonlinear system of equations is simplified into the solution of a single nonlinear equation. To demonstrate the effectiveness of the proposed approach, simulations are conducted to evaluate the efficiency of the algorithm. The simulations conducted in this study indicate that the proposed inverse kinematic (IK) algorithm improves computational speed by a significant margin. Specifically, it achieves a speedup of 2.8 × 10^3^ over the Levenberg–Marquardt (LM) method. In addition, experimental results demonstrate that the coupled-motion system achieves high levels of accuracy. Specifically, the repetitive positioning accuracy is measured to be 0.9 mm, and the tracking accuracy is 1.5 mm. This paper is significant for dealing with the coupling of the compound continuum robot.

## 1. Introduction

Robotic-assisted minimally invasive surgery has emerged as a critical research direction in recent years, owing to its numerous benefits such as reduced tissue trauma, reduced pain, and faster postoperative recovery time [1,2]. In particular, skull base surgery poses considerable challenges in terms of safety and precision. Minimally invasive intracranial surgery, in comparison to traditional cranial surgery, involves minimal incisions and blood loss, reducing the risk of complications. However, conventional minimally invasive surgical robots are limited by their rigid internal structures, making it challenging to operate on deep intracranial lesions due to the complex nature of intracranial structures.

The continuum robot, with its small size and passive flexibility, enables good motion flexibility in constrained spaces [3,4,5]. Its application in intracranial surgery allows for accurate lesion targeting and effective reduction of robotic-arm-induced tissue damage during surgery. Among the different types of continuum robots, the concentric tube robot [6,7,8] and the cable-driven continuum robot [9,10] have emerged as popular research subjects.

The concentric tube robot, which is a type of continuum robot with small size, consists of nested tubes with pre-bending. By controlling the rotation and feed of the inner and outer tubes, the shape and end position of the concentric tube robot can be changed to achieve specific motions. Researchers have proposed various applications of the concentric tube robot. Burgner et al. [11] demonstrated the feasibility of applying the concentric tube robot in transnasal tract skull base surgery by conducting initial experiments on human cadavers. Gafford et al. [12] proposed a concentric tube robotic system for clearing central airway obstruction and showed its effectiveness in reducing complications during surgery through cadaveric experiments. Wang et al. [13] proposed a three-arm concentric tube robotic system for nasopharyngeal carcinoma surgery and demonstrated its effectiveness and feasibility through tissue resection experiments in a cranial model. These studies have optimized minimally invasive surgical robots by leveraging the advantages of small-sized concentric tube robots. However, due to its inflexible bending, the concentric tube robot also has the disadvantage of being limited by localized free space.

The cable-driven continuum robot stands out for its high flexibility and a large inter-internal cavity, offering ample space to allow the passage of surgical instruments. Murphy et al. [14] presented a cable-driven dexterous manipulator that features a large, open lumen. Based on computer-aided design (CAD) tools, simulation tests were conducted to demonstrate the manipulator’s extensive working space. Fichera et al. [15] proposed a miniature robotic endoscope with a notched distal tip that can be controllably curved using a thin tendon, achieving a high visual coverage in a phantom experiment. Wu et al. [16] developed a kinematic model of a cable-driven continuum robot based on MATLAB software, providing an effective reflection of the robot’s dynamic characteristics. With its high flexibility, the cable-driven continuum robot can be adapted to a wide variety of surgical scenarios. Nonetheless, the cable-driven mechanism of this robot limits its diameter to millimeters and presents significant challenges in further reducing its size.

With the growing interest in continuum robots, some researchers have explored combining different types of continuum robots to achieve improved performance. In 2017, Li [17] proposed the initial concept of a hybrid-driven continuum robot by combining the cable-driven continuum robot with the concentric tube robot and developed a kinematic model. He evaluated the dexterity of different configurations of the hybrid continuum, which yielded valuable insights. Subsequently, several different hybrid-driven continuum robots were proposed. Abdel-Nasser et al. [18] proposed to combine the articulated continuum robot with the concentric tube robot for use in minimally invasive surgery applications, resulting in a robot with a larger working space and enhanced dexterity. Zhang et al. [19] proposed a hybrid-driven continuum robot, which is tendon-driven and magnetic-driven in mode and achieved large-angle steering and high-precision micromanipulation simultaneously. Zhang et al. [20] proposed to combine the concentric tube robot with the notched continuum robot for minimally invasive surgery applications. They leveraged the small size of the concentric tube robot to pass through the internal hole of the notched continuum robot, resulting in a compact and flexible robot arm with a large cavity for efficient surgical instrument passage.

The combined use of the concentric tube robot with the cable-driven continuum robot represents a promising approach that capitalizes on the small size of the former alongside the variable curvature of the latter, which can enhance the overall dexterity of the system. Nonetheless, the coupling effect between these two types of continuum robots poses a significant challenge that cannot be ignored. Indeed, the coupling effect is often viewed as detrimental, as it can increase the complexity of the model and lead to large errors in the theoretical model. Falkenhahn et al. [21] employed decoupling strategies to achieve simulation results that matched the measured values after modeling the dynamics of the continuum robot. Others have endeavored to reduce the influence of coupling effects in different ways. Roy et al. [22] modified the hydrostatic model to mitigate such effects, while Bryson et al. [23] established coupling boundaries in order to improve kinematic model accuracy. Through these efforts, researchers have sought to either avoid coupling effects altogether or reduce the errors resulting from coupling.

Based on past research on the compound continuum robot, this paper attempts to approach the coupling from a new perspective. The relative positions of the concentric tube and the cable-driven continuum are changed to control the robot by using the coupled motion of the two. Based on the assumption of piecewise constant curvature [24], a kinematic model of the compound continuum robot is designed. 

These contributions can be summarized as follows:A new idea to deal with coupling is proposed, which is to use coupling motion to achieve surgical operation. And it is experimentally verified that this new way has higher control accuracy compared with the robot that drives the continuum only by cables.A polynomial-curve-fitting-based inverse kinematic algorithm for the compound continuum robot is designed. Simulations show the algorithm has a good performance in terms of accuracy and computational time.Optimization of the polynomial fitting curve in the inverse kinematic algorithm using experimental data improved the solution accuracy by a factor of 3.5.

The rest of this paper is as follows. Section 2 describes the structure of the compound continuum robot, including some necessary structural parameters, and the way to control the robot’s motion. Section 3 is about the coupled-motion analysis of the compound continuum robot. Section 4 is the kinematic modeling of the compound continuum robot, including the positive kinematics and inverse kinematics and its solution algorithm. In Section 5, the established kinematic model is simulated to verify the feasibility and efficiency of the algorithm. Section 6 describes the experimental programs and their results. Section 7 provides a discussion on the relevant studies. Section 8 concludes the whole paper.

## 2. Structure of the Compound Continuum Robot

This paper presents the structure of a compound continuum robot, which is illustrated in Figure 1. The robot has a flexible cable-driven continuum exterior, and its joints are connected by four structural cables that are distributed at intervals of 90°. The use of structural cables eliminates the presence of discrete joints in the robot. Control of the continuum’s shape is achieved by the four drive cables that are also distributed at 90° intervals. Adjacent structural cables are positioned 45° from the drive cables, and the two opposite drive cables are grouped together, enabling each group of drive cables to control the cable-driven continuum’s bending in two directions within one plane. Simultaneous operation of four cables allows for bending in multiple directions. The cable-driven continuum’s rotation angle is regulated by the motor, enabling circular rotation. The internal hole of the robot serves as a movement channel for a concentric tube with pre-bending. Motor control enables the circumferential rotation and back-and-forth feeding of the concentric tube, with the feeding range restricted inside the cable-driven continuum to enhance coupling. The internal hole of the concentric tube can be used as a surgical instrument channel. The robot can be simplified into two parts: the compound part and the cable-driven continuum part, as shown in Figure 1.

The structure parameters of the robot are shown in Table 1, where *D*_1_ is the outer diameter of the concentric tube, *d*_2_ is the inner diameter of the cable-driven continuum, and *S* is the length of the concentric tube with non-zero curvature, as shown in Figure 1.

## 3. Coupling of the Compound Continuum Robot

Simultaneous motion of the concentric tube and the cable-driven continuum results in a coupled motion. Due to the considerably larger stiffness of the concentric tube relative to that of the cable-driven continuum, the latter’s influence on the attitude of the former is negligible. Therefore, only the impact of the concentric tube’s motion on the attitude of the cable-driven continuum is taken into account.

The initial form of the cable-driven continuum is determined by the driver cables, as shown in Figure 2. Initially, the lengths of the four driver cables are *L*_1_, *L*_2_, *L*_3_, and *L*_4_, respectively, where *L*_1_ > *L*_2_ = *L*_3_ > *L*_4_. The curvature of the concentric tube is *k*. The coupled motion of the two can be described as follows: the concentric tube feeds a length s in the cable-driven continuum; meanwhile, the bending angle of the compound part is *sk*. And then, the concentric tube rotates an angle *ϕ*_1_; meanwhile, the lengths of the four driver cables of the compound part are *s*_1_, *s*_2_, *s*_3_, and *s*_4_, where
(1)$$\{\begin{array}{c} s_{1}=s+\left(skD\cos\phi_{1}\right)/2\\ s_{2}=s-\left(skD\sin\phi_{1}\right)/2\\ s_{3}=s+\left(skD\sin\phi_{1}\right)/2\\ s_{4}=s-\left(skD\cos\phi_{1}\right)/2 \end{array}$$

Assuming that the stretching of the driver cables can be disregarded, their lengths are considered to be constant. The driver cables’ lengths in the cable-driven continuum part are given by (*L*_1_ − *s*_1_), (*L*_2_ − *s*_2_), (*L*_3_ − *s*_3_), and (*L*_4_ − *s*_4_), respectively. These lengths satisfy the following relation:(2)$$\left\{ \begin{array}{l} \frac{L_{1}-s_{1}}{\theta }-\frac{D\cos\phi_{2}}{2}=\frac{L-s}{\theta }\\ \frac{L_{2}-s_{2}}{\theta }-\frac{D\sin\phi_{2}}{2}=\frac{L-s}{\theta } \end{array}\right.$$
where *θ* is the bending angle of the cable-driven continuum part, *ϕ*_2_ is the rotation angle of the plane where the bending direction of the cable-driven continuum part is located, and *L* is the total length of the robot, as shown in Figure 2.

Combining (1) and (2), it can obtain that:(3)$$\phi_{2}=\arctan\frac{2L_{2}+skD\sin\phi_{1}-2L}{2L_{1}-skD\cos\phi_{1}-2L}$$
(4)$$\theta =\frac{2L_{1}-skD\cos\phi_{1}-2L}{D\cos\phi_{2}}$$

## 4. Kinematic Model

In this paper, a kinematic model is developed under the assumption of constant curvature. In contrast to previous studies, the coupling effect between the concentric tube and the cable-driven continuum is utilized to accomplish precise manipulation of the robot. To exploit this coupling effect effectively, the range of motion of the concentric tube is confined within the cable-driven continuum. Due to the significant stiffness of the concentric tube, the influence of the cable-driven continuum on the stiffness of the concentric tube is negligible. 

In Figure 3, the *O*_1_*O*_2_ section represents the compound part, while the *O*_2_*P*_E_ section represents the cable-driven continuum part. The total length of the robot is denoted by *L*, and the concentric tube is characterized by a preset curvature *k*. During operation, the concentric tube feeds a length *s* into the cable-driven continuum and then undergoes rotation by an angle *ϕ*_1_. As presented in Figure 3, the *O*_1_*O*_2_ section has a length *s*, a rotation angle *ϕ*_1_, and a bending angle *sk*. On the other hand, the length of the *O*_2_*P*_E_ section is *L* − *s*, with the deflection angle around the *Z*_2_-axis as *ϕ*_2_, and the bending angle being *θ*.

Traditional methods generally use the Denavit–Hartenberg (DH) method to establish the kinematic model and then solve the inverse kinematics using the Levenberg–Marquardt (LM) method [25,26]. The LM method obtains the result through a series of matrix iterations and is widely used. For the compound continuum robot described in this paper, after establishing the kinematic model using the DH method, the LM method is used to solve the matrix equation containing the variables *s*, *ϕ*_1_, *ϕ*_2_, and *θ*. However, this method has a slow solving speed. In this paper, a new kinematic model is established, and a new inverse kinematics algorithm is designed. Through simple analysis, a system of equations to be solved is obtained. By using polynomial fitting, the solution of a system of equations is transformed into the solution of an equation with two variables. Then, the Newton iteration method is used to quickly solve it. This algorithm greatly improves the solving speed while ensuring the accuracy of the solution. 

### 4.1. Forward Kinematics Model

When *s*, *ϕ*_1_, *ϕ*_2_, and *θ* are all known quantities, the position of *P*_E_ on the world coordinate system *O*_1_-*X*_1_*Y*_1_*Z*_1_ is as follows:(5)$$P_{E}^{O_{1}}=\left[\begin{array}{c} x\\ y\\ z \end{array}\right]$$
where *x*, *y*, and *z* are functions with respect to *L*, *k*, *s*, *ϕ*_1_, *ϕ*_2_, and *θ*. The expressions of *x*, *y*, and *z* can be derived by the following matrix transformation analysis.

As shown in Figure 3, the position of *P*_E_ of the compound continuum robot on the relative coordinate system *O*_2_-*X*_2_*Y*_2_*Z*_2_ is as follows:(6)$$P_{E}^{O_{2}}=\left[\begin{array}{c} x_{2}\\ y_{2}\\ z_{2} \end{array}\right]=\left[\begin{array}{c} \left(L-s\right)\left(1-\cos\theta \right)\cos\phi_{2}/\theta \\ \left(L-s\right)\left(1-\cos\theta \right)\sin\phi_{2}/\theta \\ \left(L-s\right)\sin\theta /\theta \end{array}\right]$$

The position of *P*_E_ on *O*_1_-*X*_1_*Y*_1_*Z*_1_ is $P_{E}^{O_{1}}$ and on *O*_2_-*X*_2_*Y*_2_*Z*_2_ is $P_{E}^{O_{2}}$. They include the following relationship:(7)$$\left[\begin{array}{c} P_{E}^{O_{1}}\\ 1 \end{array}\right]=\left[\begin{array}{cc} R_{O_{2}}^{O_{1}} & P_{O_{2}}^{O_{1}}\\ 0 & 1 \end{array}\right]\cdot \left[\begin{array}{c} P_{E}^{O_{2}}\\ 1 \end{array}\right]=\left[\begin{array}{c} R_{O_{2}}^{O_{1}}\cdot P_{E}^{O_{2}}+P_{O_{2}}^{O_{1}}\\ 1 \end{array}\right]$$
where $R_{O_{2}}^{O_{1}}$ and $P_{O_{2}}^{O_{1}}$ are the rotation and translation matrices relative to the coordinate system *O*_2_-*X*_2_*Y*_2_*Z*_2_, respectively. From this, the expressions of (5) with respect to *L*, *k*, *s*, *ϕ*_1_, *ϕ*_2_, and *θ* can be obtained as follows:(8)$$P_{E}^{O_{1}}=\left[\begin{array}{c} x\\ y\\ z \end{array}\right]=\left[\begin{array}{c} \left(L-s\right)\left[\left(1-c\;\theta \right)\left(c\;\phi_{1}c\;\phi_{2}c\;sk-s\;\phi_{1}s\;\phi_{2}\right)+c\;\phi_{1}s\;sks\;\theta \right]/\theta +\left(\left(1-c\;sk\right)c\;\phi_{1}\right)/k\\ \left(L-s\right)\left[\left(1-c\;\theta \right)\left(s\;\phi_{1}c\;\phi_{2}c\;sk+c\;\phi_{1}s\;\phi_{2}\right)+s\;\phi_{1}s\;sks\;\theta \right]/\theta +\left(\left(1-c\;sk\right)c\;\phi_{1}\right)/k\\ \left(L-s\right)\left[c\;sk\;s\;\theta -s\;sk\;c\;\phi_{2}\left(1-c\;\theta \right)\right]/\theta +\left(s\;sk\right)/k \end{array}\right]$$
where c *sk* = cos *sk*, s *sk* = sin *sk*, c *θ* = cos *θ*, s *θ* = sin *θ*, c *ϕ*_1_ = cos *ϕ*_1_, s *ϕ*_1_ = sin *ϕ*_1_, c *ϕ*_2_ = cos *ϕ*_2_, and c *ϕ*_2_ = cos *ϕ*_2_, which are the same as below.

Substituting (3) and (4) into (8) yields the coordinates of *P*_E_ versus (*s*, *ϕ*_1_). From this, the motion space when controlling the robot using coupled motion can be obtained, as shown in Figure 4. The relevant parameters of this simulation experiment are shown in Table 2. It can be seen that it is feasible to control the motion of the robot by the coupling.

### 4.2. Inverse Kinematics and Its Algorithm

The process for obtaining the inverse kinematic solution of the robot is illustrated in the flowchart presented in Figure 5.

The following is a detailed description of Figure 5.

As shown in Figure 3, the position of *P*_E_ on the world coordinate system *O*_1_-*X*_1_*Y*_1_*Z*_1_ is expressed by (5), and when *x*, *y*, and *z* are known quantities in (5), then (*s*, *ϕ*_1_, *ϕ*_2_, *θ*) can be obtained from the inverse kinematic analysis.

From Equation (6), it is easy to obtain
(9)$$\phi_{2}=\arctan\frac{y_{2}}{x_{2}}$$

From the geometric relationship, it is easy to obtain
(10)$$\theta =2\arctan\frac{\sqrt{x_{2}{}^{2}+y_{2}{}^{2}}}{z_{2}}$$

As shown in Figure 3, the position of *P*_M_ of the compound part on the world coordinate system *O*_1_-*X*_1_*Y*_1_*Z*_1_ can be expressed as
(11)$$P_{M}=\left[\begin{array}{c} x_{1}\\ y_{1}\\ z_{1} \end{array}\right]=\left[\begin{array}{c} \left(1-\cos sk\right)\cos\phi_{1}/k\\ \left(1-\cos sk\right)\sin\phi_{1}/k\\ \sin sk/k \end{array}\right]$$

From (11), the following can be obtained:(12)$$\phi_{1}=\arctan\frac{y_{1}}{x_{1}}$$
(13)$$y_{1}=\{\begin{array}{c} \sqrt{{\left(1-\cos sk\right)}^{2}-k^{2}x_{1}{}^{2}}/k,y_{1}\geq 0\\ -\sqrt{{\left(1-\cos sk\right)}^{2}-k^{2}x_{1}{}^{2}}/k,y_{1}<0 \end{array}$$
(14)$$z_{1}=\sin sk/k$$

When $\phi_{1}\in \left[0,\pi \right]$, *y*_1_ ≥ 0, and when $\phi_{1}\in \left(\pi ,2\pi \right)$, *y*_1_ < 0. Due to the symmetry of the two cases in (13), the analysis in the case at *y*_1_ ≥ 0 is enough.

With (12) and (13), the rotation matrix $R_{O_{2}}^{O_{1}}$ can be expressed as
(15)$$\begin{array}{cc} R_{O_{2}}^{O_{1}} & =\left[\begin{array}{ccc} \left(x_{1}\;c\;sk\right)/m & -y_{1}/m & \left(x_{1}\;s\;sk\right)/m\\ \left(y_{1}\;c\;sk\right)/m & x_{1}/m & \left(y_{1}\;s\;sk\right)/m\\ -s\;sk & 0 & c\;sk \end{array}\right]\\ & =\left[\begin{array}{ccc} \left(kx_{1}\;c\;sk\right)/n & -ky_{1}/n & \left(kx_{1}\;s\;sk\right)/n\\ \left(ky_{1}\;c\;sk\right)/n & kx_{1}/n & \left(ky_{1}\;s\;sk\right)/n\\ -s\;sk & 0 & c\;sk \end{array}\right] \end{array}$$
where $m=\sqrt{x_{1}^{2}+y_{1}^{2}}$, n=1−c sk.

Combining (5), (11), and (15), the results can be obtained as
(16)$$\begin{array}{cc} P_{E}^{O_{2}} & =R_{O_{2}}^{O_{1}}{}^{-1}\left(P_{E}^{O_{1}}-P_{M}\right)\\ & =\left[\begin{array}{c} \left(z_{1}-z\right)s\;sk-\left(x_{1}{}^{2}+y_{1}{}^{2}-x_{1}x-y_{1}y\right)\left(1-c\;sk\right)c\;sk/\left(k\left(x_{1}{}^{2}+y_{1}{}^{2}\right)\right)\\ \left(x_{1}y-xy_{1}\right)\left(1-c\;sk\right)/\left(k\left(x_{1}{}^{2}+y_{1}{}^{2}\right)\right)\\ \left(z-z_{1}\right)c\;sk-\left(x_{1}{}^{2}+y_{1}{}^{2}-x_{1}x-y_{1}y\right)\left(1-c\;sk\right)s\;sk/\left(k\left(x_{1}{}^{2}+y_{1}{}^{2}\right)\right) \end{array}\right] \end{array}$$

Combining (13), (14), and (16), thus eliminating *y*_1_ and *z*_1_ in (16), the result can be obtained as
(17)$$\left[\begin{array}{c} x_{2}\\ y_{2}\\ z_{2} \end{array}\right]=\left[\begin{array}{c} \left(s^{2}\;sk\right)/k-zs\;sk-\left(\left(1-c\;sk\right)/k-\left(kx_{1}x+y\sqrt{{\left(1-c\;sk\right)}^{2}-k^{2}x_{1}{}^{2}}\right)/\left(1-c\;sk\right)\right)c\;sk\\ \left(kx_{1}y-x\sqrt{{\left(1-c\;sk\right)}^{2}-k^{2}x_{1}{}^{2}}\right)/\left(1-c\;sk\right)\\ zc\;sk-\left(s{}^{2}sk\right)/\left(2k\right)-\left(\left(1-c\;sk\right)/k-\left(kx_{1}x+y\sqrt{{\left(1-c\;sk\right)}^{2}-k^{2}x_{1}{}^{2}}\right)/\left(1-c\;sk\right)\right)s\;sk \end{array}\right]$$

From (6), the workspace of *P*_E_ on the relative coordinate system *O*_2_-*X*_2_*Y*_2_*Z*_2_ is a surface formed by rotation around the *Z*_2_-axis, and this surface can be expressed by the implicit equation:(18)$$F\left(x_{2},y_{2},z_{2}\right)=0$$

From (6), the following can be obtained:(19)$$\sqrt{x_{2}{}^{2}+y_{2}{}^{2}}=\left(L-s\right)\left(1-c\;\theta \right)/\theta$$
(20)$$z_{2}=\left(L-s\right)s\;\theta /\theta$$

From (19) and (20), it can be obtained that the relationship between $\sqrt{x_{2}{}^{2}+y_{2}{}^{2}}$ and *z*_2_ is independent of *ϕ*_2_, so the implicit equation on the coordinate system *R*-*O*-*Z*_2_ can be established as
(21)$$G\left(r,z_{2}\right)=0$$
where $r=\sqrt{x_{2}{}^{2}+y_{2}{}^{2}}$. Since this implicit equation is difficult to solve, it is necessary to choose a simple curve to replace this equation. When s∈[0,L], this curve varies with *s*. This curve is related to the position of $P_{E}^{O_{2}}$, and it is equivalent to the end position curve of the cable-driven continuum robot of length (*L* − *s*) in bending motion in any plane. The curve can be fitted as
(22)$$w=a_{0}+a_{1}t+a_{2}t^{2}+a_{3}t^{3}+a_{4}t^{4}$$
where
(23)t=(L−s)(1−c θ)/θ
(24)w=(L−s)s θ/θ*a_i_* is the coefficient obtained from the fit, which is obtained by solving the following equation:(25)$${\sum_{i=1}\left(a_{0}+a_{1}t_{i}+a_{2}t_{i}{}^{2}+a_{3}t_{i}{}^{3}+a_{4}t_{i}{}^{4}-w_{i}\right)}^{2}=0$$
where
(26)$$t_{i}=\left(L-s\right)\left(1-c\;\theta_{i}\right)/\theta_{i}$$
(27)$$w_{i}=\left(L-s\right)s\;\theta_{i}/\theta_{i}$$
where $\theta_{0}=0$, $\theta_{i}=\theta_{i-1}+\frac{\pi i}{2N}$, and *N* is a sufficiently large positive integer. From this, the implicit equation for *x*_1_ and *s* can be obtained as
(28)$$z_{2}=a_{0}+a_{1}r+a_{2}r^{2}+a_{3}r^{3}+a_{4}r^{4}$$
where $r=\sqrt{x_{2}{}^{2}+y_{2}{}^{2}}$. And *x*_2_, *y*_2_, and *z*_2_ are defined by (17).

Each substitution of a value of *s* is re-fitted, and this method is obviously very tedious. Therefore, (22) is changed to
(29)$$W=A_{0}+A_{1}T+A_{2}T^{2}+A_{3}T^{3}+A_{4}T^{4}$$
where
(30)T=Lt/(L−s)=L(1−c θ)/θ
(31)W=Lw/(L−s)=Ls θ/θ*A_i_* is the coefficient obtained from the fit, which is obtained by solving the following equation:(32)$${\sum_{i=1}\left(A_{0}+A_{1}T_{i}+A_{2}T_{i}{}^{2}+A_{3}T_{i}{}^{3}+A_{4}T_{i}{}^{4}-W_{i}\right)}^{2}=0$$
where
(33)$$T_{i}=\sqrt{x_{i}{}^{2}+y_{i}{}^{2}}$$
(34)$$W_{i}=z_{i}$$

The fitting coefficient is solved as
(35)$$\left[\begin{array}{ccccc} A_{0} & A_{1} & A_{2} & A_{3} & A_{4} \end{array}\right]=\left[\begin{array}{ccccc} 89.85 & 0.064 & -0.014 & 2.19\times 10^{-4} & -3.00\times 10^{-6} \end{array}\right]$$

However, (19) and (20) are derived based on the theoretical derivation of constant curvature, and the factors of torsion, friction, shear, and axial elongation are not considered in the modeling process. So, the fitting coefficients can easily cause large errors when solved by this method. Therefore, in this paper, the actual end coordinates are used as the fitting data by experimental method to reduce the fitting errors. The end coordinate (*x_i_*, *y_i_*, *z_i_*) of the cable-driven continuum robot with length *L* is obtained experimentally, and let *L* = 90 mm. In the experimental process, the end curve of the cable-driven continuum robot is changeable when it is gradually bent and when it is gradually straightened due to the change of friction direction, so two sets of fitting coefficients are obtained.

When the cable-driven continuum robot is gradually bent, the fitting coefficient is as follows:(36)$$\left[\begin{array}{ccccc} A_{0} & A_{1} & A_{2} & A_{3} & A_{4} \end{array}\right]=\left[\begin{array}{ccccc} 90.0 & 0.1 & -0.02 & 3.1\times 10^{-4} & -3.0\times 10^{-6} \end{array}\right]$$

When the cable-driven continuum robot is gradually straightened, the fitting coefficient is as follows:(37)$$\left[\begin{array}{ccccc} A_{0} & A_{1} & A_{2} & A_{3} & A_{4} \end{array}\right]=\left[\begin{array}{ccccc} 91.4 & -0.2 & -0.03 & 5.3\times 10^{-4} & -4.3\times 10^{-6} \end{array}\right]$$

From this, the implicit equation for *x*_1_ and *s* can be obtained as
(38)$$z_{2}=A_{0}+A_{1}r+A_{2}r^{2}+A_{3}r^{3}+A_{4}r^{4}$$
where $r=\sqrt{x_{2}{}^{2}+y_{2}{}^{2}}$. And *x*_2_, *y*_2_, and *z*_2_ are defined by (17). By inputting the value of *s*, the following equation can be obtained:(39)$$F\left(x_{1}\right)=A_{0}+A_{1}r+A_{2}r^{2}+A_{3}r^{3}+A_{4}r^{4}-z_{2}=0$$

The multiset (*s*, *x*_1_) can be obtained by Newton’s iteration method. The *s* and *x*_1_ obtained from the solution are input into (13) and (12) in turn to obtain the rotation angle *ϕ*_1_ of the compound part, and the coordinate values of $P_{E}^{O_{2}}$ are obtained by inputting the solutions into (17); then, *ϕ*_2_ and *θ* are obtained from (9) and (10), respectively.

## 5. Simulation and Analysis

To analyze the performance of the algorithm for finding the numerical solution of the inverse kinematics of the compound continuum robot, two simulations are performed with the experimental environment MATLAB 2020a. Due to the symmetry of the robot workspace, the analysis in the fourth quadrant of the workspace is enough. The parameters set for the simulations are shown in Table 3, where *ε* is the iteration error and *m* is the maximum number of iterations.

### 5.1. Simulation 1

An endpoint in the workspace corresponds to multiple poses of the robot. The endpoint is fixed as *P*_E_ (31,45,60), and the multiple sets (*s*, *ϕ*_1_, *ϕ*_2_, *θ*) are solved by the inverse kinematic algorithm in Section 4, so that each point on the robot’s poses curve can be obtained. And the partial poses of the robot are shown in Figure 6.

The simulation errors in the *x*-axis, *y*-axis, and *z*-axis directions and the distances are shown in Figure 7. And the average errors are shown in Table 4.

The results are compared with the LM algorithm as shown in Table 5. The average number of iterations and the average operation time using the algorithm described in this paper are *n*_1_ and *t*_1_, respectively, and the average number of iterations and the total operation time using the LM algorithm are *n*_2_ and *t*_2_, respectively. *N* is the number of solutions.

In Simulation 1, the computational efficiency of the algorithm described in this paper is approximately 1.9 × 10^3^ times that of the LM algorithm.

### 5.2. Simulation 2

When the end of the robot moves along the curve *L*_0_, the partial poses of the robot corresponding to each of the 26 equidistant points are found, as shown in Figure 8.

The simulation errors in the *x*-axis, *y*-axis, and *z*-axis directions and the distance are shown in Figure 9.

The average errors in the *x*-axis, *y*-axis, and *z*-axis directions and the distance are shown in Table 6.

The results are compared with the LM algorithm as shown in Table 7. The average number of iterations and the average operation time using the algorithm described in this paper are *n*_1_ and *t*_1_, respectively, and the average number of iterations and the total operation time using the LM algorithm are *n*_2_ and *t*_2_, respectively. *N* is the number of solutions.

In Simulation 2, the computational efficiency of the algorithm described in this paper is approximately 2.8 × 10^3^ times that of the LM algorithm.

It can be seen that, compared with the LM algorithm, the computational efficiency of the algorithm in this paper has obvious advantages, and the greater the number of solutions, the more obvious this advantage is.

## 6. Experiment and Analysis

The physical prototype and experimental platform used for the compound continuum robot are depicted in Figure 10. Specifically, the external component of the robot is a photosensitive resin notched tube, while the internal concentric tube is a 304 stainless steel round tube. These components have the same dimensions and parameters as those specified in Table 1. The driver cables utilized are comprised of nickel–titanium alloy cables with a diameter of 0.2 mm. During experimentation, magnetic tracking is implemented for end position tracking, while signal acquisition and output is achieved through the use of an EIMO sensor.

### 6.1. Optimization Effect of the Fitted Curve

The constant curvature assumption of the robot does not hold exactly in practice, mainly due to the varied contact patterns observed between different joints of the cable-driven continuum in various poses. To address this issue, this paper employs an optimization approach in conjunction with experimental data to minimize errors arising from this source. The fitted curves before and after optimization have been provided in Section 4.2, and this experiment aims to validate the optimization performance. Specifically, with the feed *s* of the concentric tube set at a constant value of 31 mm, the concentric tube is subjected to arbitrary angle rotations *ϕ*_1_, with the resulting end position coordinates recorded. The rotation angle *ϕ*_1_ is then computed using the algorithm described in Section 4.2. This process is repeated 10 times, both before and after optimization, with the drive parameter *ϕ*_1_ errors depicted in Figure 11.

As shown in Table 8, the optimization effectively reduces errors. After optimization, there are still errors due to the fact that the established kinematics ignore the length of the driver cables being stretched.

### 6.2. Accuracy Analysis of Coupled Motion

In this section, the tracking accuracy of the continuum under coupling is compared with that of the normal cable-driven continuum robot. Some pictures of the experiment are shown in Figure 12.

The theoretical motion trajectory of the robot’s endpoint controlled by coupling can be derived using the approach presented in Section 4.1. During the experiment, the feed and rotation of the concentric tube are controlled by maintaining the driver cables’ lengths at fixed values, and the resulting endpoint motion trajectory is determined. Specifically, the driver cables’ lengths are kept constant at *L*, with other relevant parameters being provided in Table 9. Under these conditions, the concentric tube is rotated one full circle, and the experimental and theoretical trajectories are illustrated in Figure 13a. Additionally, the errors between the actual and theoretical coordinates are presented in Figure 13b.

The endpoint (*x*_c_, *y*_c_, *z*_c_) is obtained from (40) based on the assumption of constant curvature when the cable-driven continuum robot makes a bending motion in any plane.
(40)$$\left[\begin{array}{c} x_{c}\\ y_{c}\\ z_{c} \end{array}\right]=\left[\begin{array}{c} L\left(1-\cos\theta_{c}\right)\cos\phi_{c}/\theta_{c}\\ L\left(1-\cos\theta_{c}\right)\sin\phi_{c}/\theta_{c}\\ L\sin\theta_{c}/\theta_{c} \end{array}\right]$$
where *ϕ*_c_ is the angle between the plane in which the robot bends and the plane *X*_c_-*O*-*Z*_c_. And *θ*_c_ is the bending angle of the robot.

In the experiment, the lengths of the four cables are made to vary as shown in (41) to drive the robot in a bending motion.
(41)$$\left\{ \begin{array}{l} \Delta L_{1}=\Delta L_{2}=5\times \mathrm{abs}\left(\sin A\right)\\ \Delta L_{3}=\Delta L_{4}=-5\times \mathrm{abs}\left(\sin A\right) \end{array}\right.$$
where *A* is a time parameter.

*θ*_c_ is expressed as
(42)$$\theta_{c}=2\sqrt{2}\left(L-L_{\min}\right)/D$$
where *L* is the length of the cable-driven continuum robot, and *D* is the diameter of the circle in which the driver cables are located, as described in Table 1 and Table 4. *L*_min_ is the length of the shortest driver cable among the four driver cables.

The position errors of the robot are shown in Figure 14.

The tracking accuracies of the compound continuum robot controlled by the coupling and the cable-driven continuum robot are shown in Table 10. 

The tracking accuracy of the robot controlled by the coupling has an obvious advantage, and it is 3.2 times more accurate than the tracking of the cable-driven continuum robot.

The kinematic model that has been developed is based on the assumption that the curvature of the continuum remains constant, and it neglects the length variations of the driver cables caused by stretching. However, the contact pattern between the continuum’s joints varies with the robot’s posture, which leads to errors. In addition, for compound continuum robots, another source of errors arises due to the non-rigid nature of the concentric tube. Namely, the motion of the cable-driven continuum induces changes in the curvature of the concentric tube, which are not accounted for in the kinematic model.

## 7. Discussion

In fact, the compound continuum robot described in this paper possesses three motion modes, namely high-stiffness mode, high-dexterity mode, and coupling mode. For the high-stiffness mode, the cable-driven continuum completely coincides with the concentric tube, such that the pre-bending direction of the concentric tube always opposes the load direction. Additionally, simultaneous control of the rotational motion of the concentric tube and the bending and rotational motion of the cable-driven continuum enables motion at the robot’s end effector. For the high-dexterity mode, simultaneous control of the feeding and rotational motion of both the concentric tube and the cable-driven continuum facilitates coordinated motion at the robot’s end effector. Research on these two motion modes is currently underway. The present study is based on the coupling mode, wherein the feeding motion range of the concentric tube is confined within the cable-driven continuum. Through the coupling effect exerted by the concentric tube on the cable-driven continuum, the shape of the latter is altered, enabling high-precision motion at the robot’s end effector. In fact, the coupling mode also encompasses utilizing the coupling effect of the cable-driven continuum on the concentric tube to modify the tube’s pre-bending angle, which is a future research direction for this study. The investigation of the coupling mode in this paper represents a significant contribution to the study of the compound continuum robot system.

## 8. Conclusions

The mutual coupling of concentric tube and cable-driven continuum components in compound continuum robots leads to reduced control accuracy, which is typically addressed through error reduction and compensation techniques. In this paper, the coupling of the compound continuum robot components is harnessed to change the end effector position of the robot. A kinematic model and an inverse kinematic (IK) algorithm are established based on assumptions of constant curvature. And the IK algorithm is reduced to the problem of solving a nonlinear equation which improves computational efficiency. By investigating the length of driver cables in relation to the robot’s poses, the robot’s coupled-motion workspace is obtained. Through simulations, the proposed IK algorithm achieves significant computation speedups over the Levenberg–Marquardt (LM) method, achieving a speedup of 1.9 × 10^3^ for 10 solutions and 2.8 × 10^3^ for 130 solutions. To improve accuracy, polynomial fitting curves in the algorithm are optimized using experimental data, leading to a 3.5-fold improvement in solution accuracy. The tracking accuracy of the coupled motion is found to be 1.5 mm, accounting for 1.6% of the total continuum length, and is 3.2 times higher than the tracking accuracy of the cable-driven continuum robot. Overall, this paper presents a novel approach for managing coupling in continuum robots.

In the future, the coupling mode compound continuum robot, due to its better accuracy and control performance, will be able to apply to various minimally invasive surgeries including procedures such as local laser ablation and radioactive particle implantation.

## Figures and Tables

**Figure 1 sensors-23-06407-f001:**
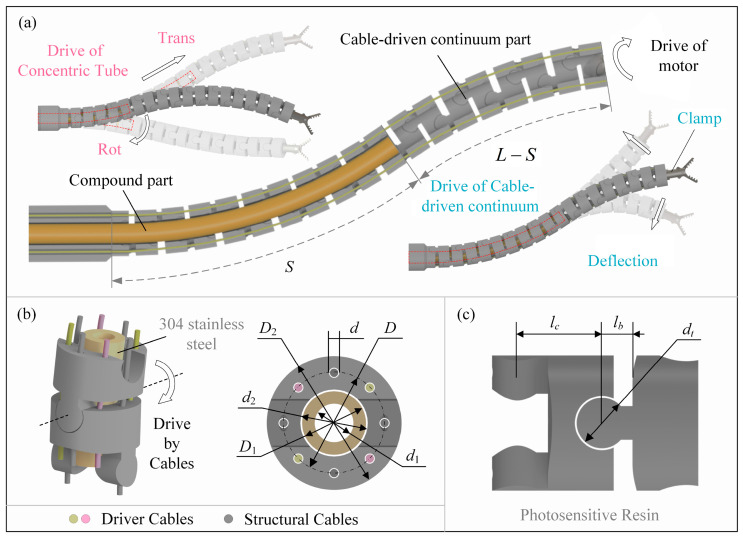
The structure of the compound continuum robot: (**a**) The overall structure, (**b**) Cable-driven mode and robot’s radial dimensions and (**c**) The axial dimensions of the robot joints.

**Figure 2 sensors-23-06407-f002:**
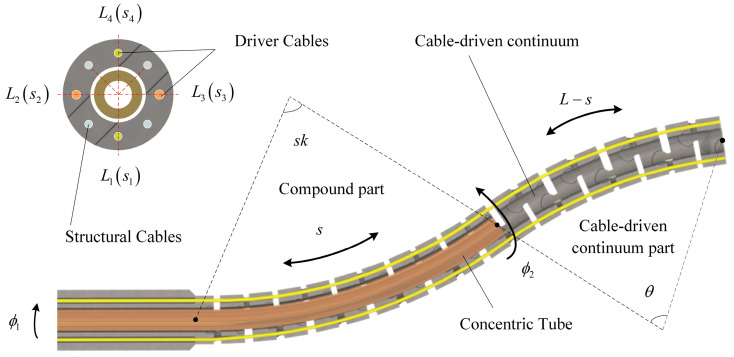
Coupled motion of the compound continuum robot.

**Figure 3 sensors-23-06407-f003:**
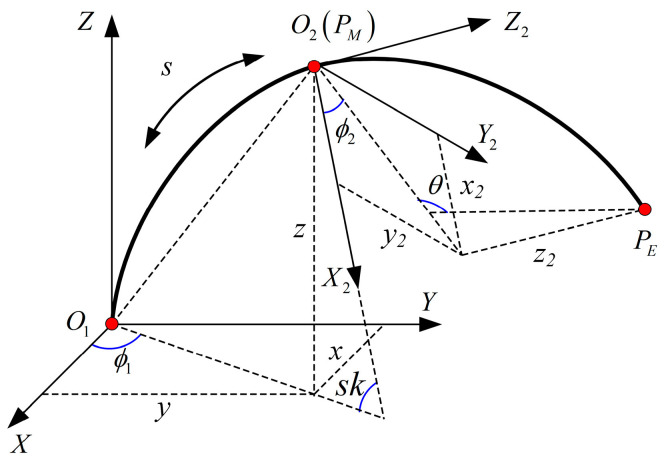
Simplified coordinate diagram of the robot.

**Figure 4 sensors-23-06407-f004:**
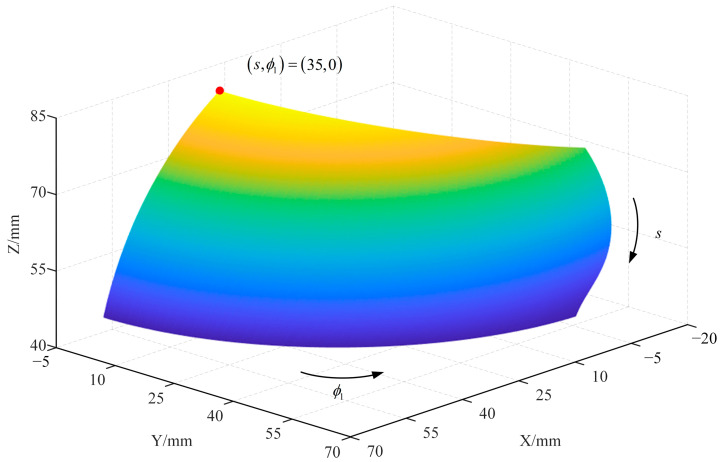
Working space of the robot: The figure illustrates the reachable workspace of the robot under the conditions described in Table 2. The two arrows in the figure indicate the direction of changes in the position of the robot’s end effector as *s* and *ϕ*_1_ increase. The red dot represents the initial position.

**Figure 5 sensors-23-06407-f005:**
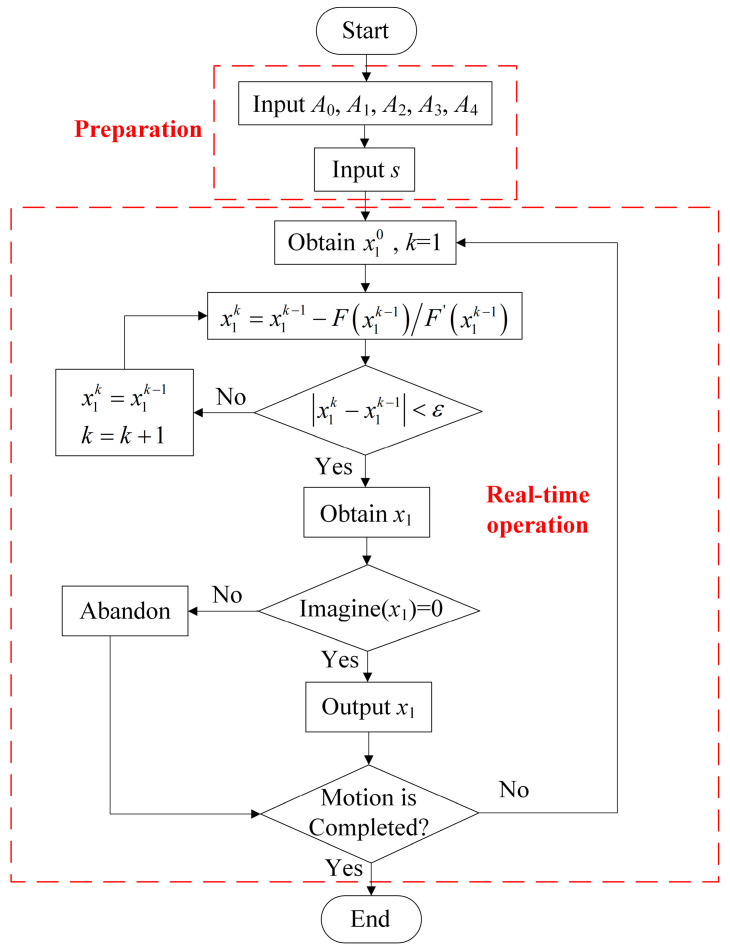
Flow chart of inverse kinematics solution: *A_i_* is the coefficient obtained from the fit, which is described in the subsequent text. *k* is the step of iteration, and $x_{1}^{k}$ represents the value of *x*_1_ after *k* iterations. *ε* is the iteration error. And *F*(*x*_1_) is a function of *x*_1_, which is defined by (39).

**Figure 6 sensors-23-06407-f006:**
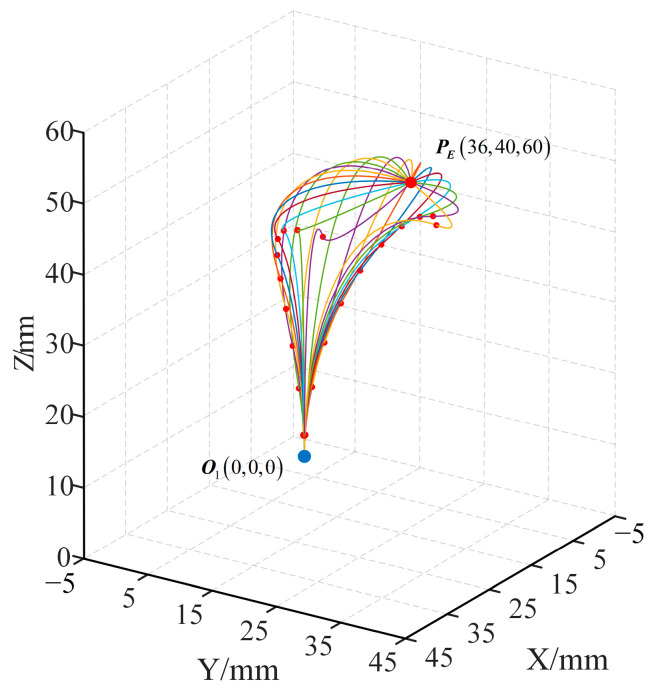
Simulation 1: the blue dot represents the robot’s proximal end, the larger red dot represents the robot’s distal end, the smaller red dots represent the distal ends of the concentric tube, and the curves represent the robot’s poses.

**Figure 7 sensors-23-06407-f007:**
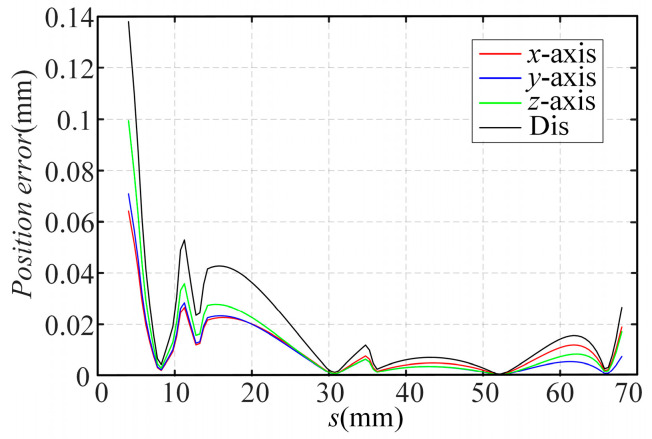
Errors in Simulation 1.

**Figure 8 sensors-23-06407-f008:**
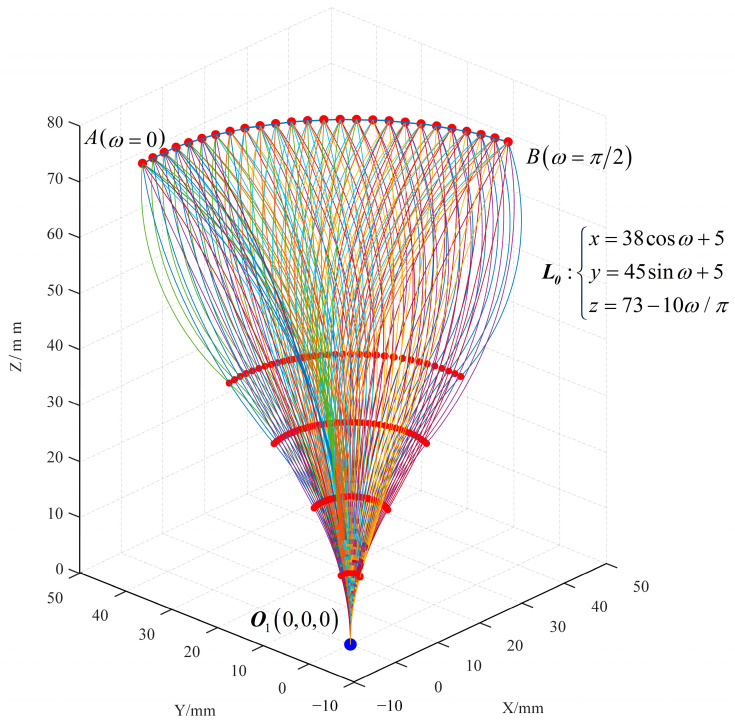
Simulation 2: the blue dot represents the robot’s proximal end, the larger red dots represent the robot’s distal ends, the smaller red dots represent the distal ends of the concentric tube, and the curves represent the robot’s poses.

**Figure 9 sensors-23-06407-f009:**
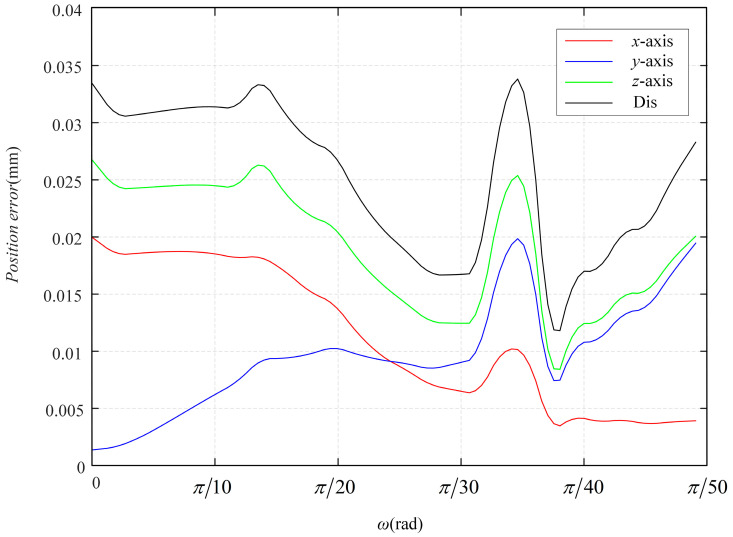
Errors in Simulation 2.

**Figure 10 sensors-23-06407-f010:**
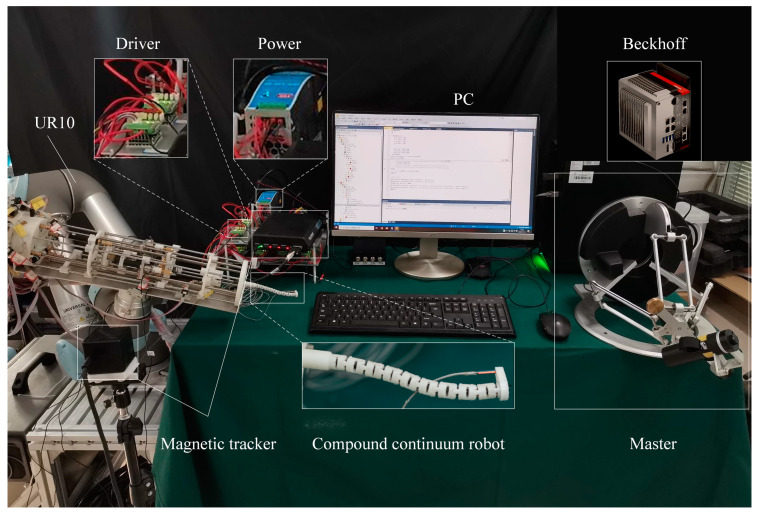
The physical prototype and experimental platform.

**Figure 11 sensors-23-06407-f011:**
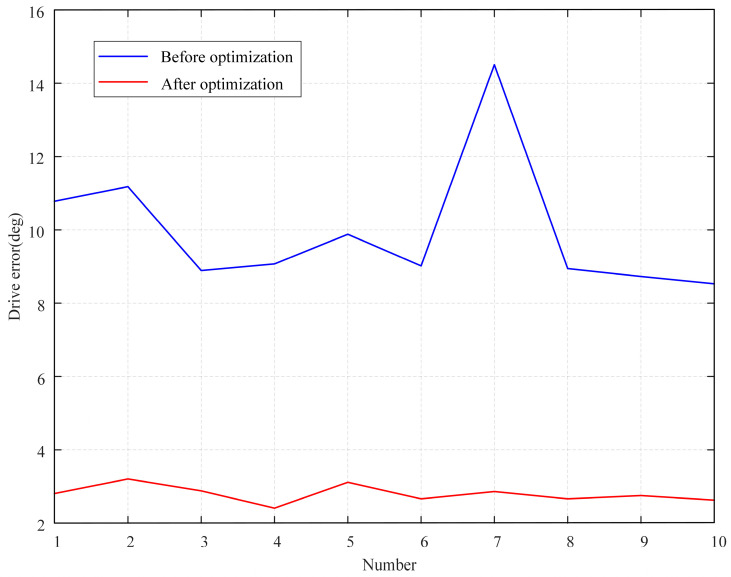
Errors of drive parameter *ϕ*_1_.

**Figure 12 sensors-23-06407-f012:**
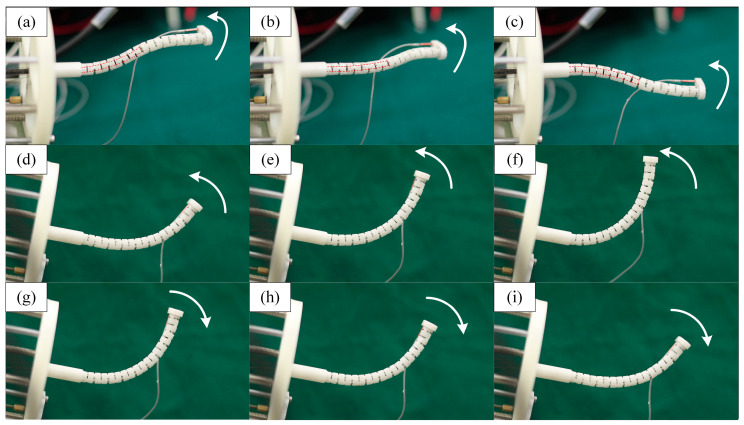
Some pictures of the experiment: (**a**–**c**) are pictures of the compound continuum robot controlled by coupling, and (**d**–**i**) are pictures of the bending motion of the cable-driven continuum robot.

**Figure 13 sensors-23-06407-f013:**
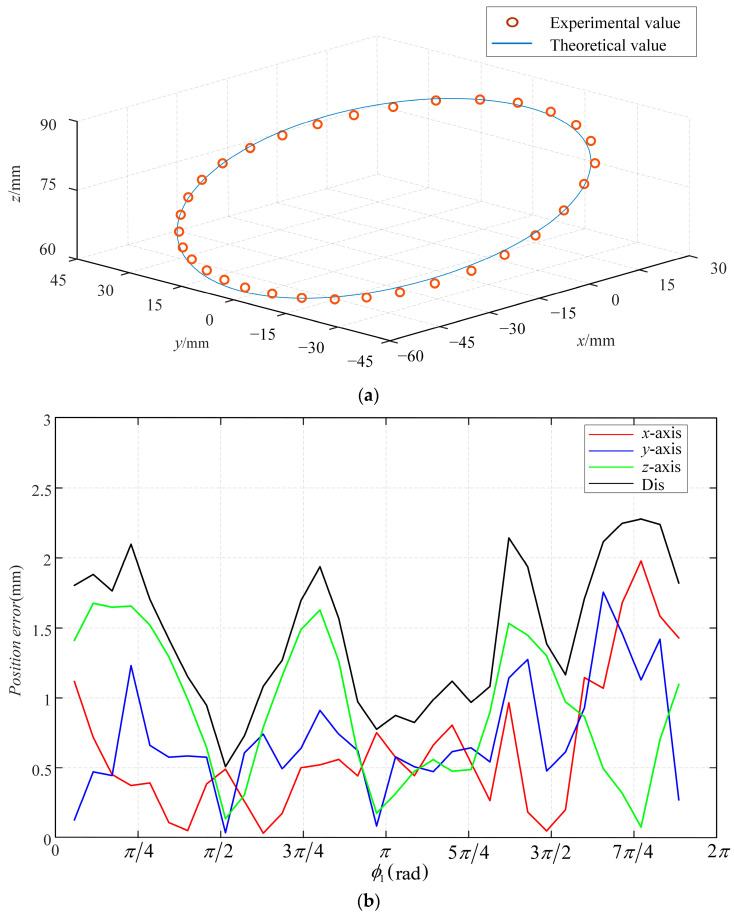
The experimental results and theoretical results: (**a**) movement track and (**b**) coordinate errors.

**Figure 14 sensors-23-06407-f014:**
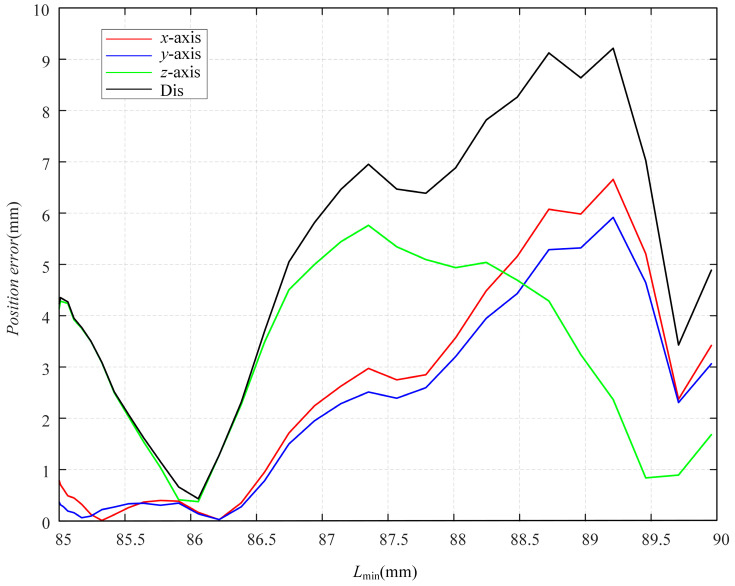
Position errors of the cable-driven continuum.

**Table 1 sensors-23-06407-t001:** Structural parameters of the robot.

Parameters	Values	Parameters	Values
*D*_1_/mm	3.5	*d*/mm	0.6
*d*_1_/mm	2.5	*D*/mm	6.9
*S*/mm	50	*δ*/mm	4
*D*_2_/mm	8.6	*l_c_*/mm	5
*d*_2_/mm	3.8	*l_b_*/mm	1.8
*L*/mm	90	*d_t_*/mm	3

**Table 2 sensors-23-06407-t002:** Parameters set for the simulation.

*L*_1_/mm	*L*_2_/mm	*L*_3_/mm	*L*_4_/mm	*k*	*s*/mm	*ϕ* _1_
90	90	90	90	π/150	From 35 to 90	From 0° to 90°

**Table 3 sensors-23-06407-t003:** Parameters set for both simulations.

Parameters	*L*/mm	*k*	*ε*	*m*	$x_{1}^{0}$
Simulation 1	90	π/150	0.01	10	0.01
Simulation 2					

**Table 4 sensors-23-06407-t004:** The average errors in Simulation 1.

Parameters	*x*/mm	*y*/mm	*z*/mm	*Dis*/mm	Dis/L×100%
Values	0.014	0.013	0.017	0.026	0.028%

**Table 5 sensors-23-06407-t005:** Evaluation of results in Simulation 1.

Parameters	*N*	*t*_1_/s	*t*_2_/s	*n* _1_	*n* _2_
Values	10	2.79 × 10^−5^	0.05418	3.7	5.5

**Table 6 sensors-23-06407-t006:** The average errors in Simulation 2.

Parameters	*x*/mm	*y*/mm	*z*/mm	*Dis*/mm	Dis/L×100%
Values	0.011	0.0096	0.019	0.025	0.028%

**Table 7 sensors-23-06407-t007:** Evaluation of results in Simulation 2.

Parameters	*N*	*t*_1_/s	*t*_2_/s	*n* _1_	*n* _2_
Values	130	1.89 × 10^−5^	0.05277	3.6	5.7

**Table 8 sensors-23-06407-t008:** The average errors of drive parameter *ϕ*_1_.

Parameters	*ϕ* _1_	*ϕ*_1_/360° × 100%
Before optimization	10.0°	2.8%
After optimization	2.8°	0.8%

**Table 9 sensors-23-06407-t009:** The average errors of drive parameter *ϕ*_1_. Experiment-related parameters.

*ϕ*_1_/rad	*L*/mm	*k*	*s*/mm
From 0 to 2π	90	π/150	31

**Table 10 sensors-23-06407-t010:** End position errors.

Parameters	*x*/mm	*y*/mm	*z*/mm	Dis/mm	Dis/*L* × 100%
Coupling-driven	0.6	0.7	0.9	1.5	1.6%
Cable-driven	2.0	1.8	3.3	4.7	5.2%

## Data Availability

Not applicable.

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
