# Peer review of "Coupling Analysis of Compound Continuum Robots for Surgery: Another Line of Thought"

_sensors, 2023, doi:10.3390/s23146407_

Round 1
Reviewer 1 Report
Section 4 The kinematic model should start on a new page 6.
Reviewer 2 Report
The concentric tube components of the continuum robot cause difficulties in modeling and control. Addressing these challenges, the present paper establishes a kinematics model, along with an inverse kinematic algorithm, considering the coupling between the components. The control algorithm can achieve faster and more precise robot control than previous strategies. The following are some suggestions that may help improve the manuscript:
-
The figures incorporated within the manuscript lack clarity and should be replaced with high-resolution alternatives to enhance their interpretability.
- The captions provided for the figures require further elaboration. Presently, they are overly concise and need to include more details to give readers an adequate understanding of the figures' content and relevance.
-
In the course of deriving the kinematics model and the inverse kinematic algorithm, it is recommended that the authors delineate more explicitly how the interplay of motion is factored in and how the proposed method differs from those presented in prior research. This clarification is particularly critical for readers unfamiliar with this field.
- Previous approaches, such as the Leven-berg-Marquardt method referred to in the manuscript, should be briefly explained and introduced. This can help readers understand what is the innovation in the newly proposed method.
The main text is clear and easy to understand.
Reviewer 3 Report
This paper leverages the coupling between the components of the compound continuum robot to accomplish specific surgical procedures.
I suggest to add a discussion section that discusses the results with the recent developed systems in the same area.
NA
Reviewer 4 Report
This work proposes a novel approach for managing coupling in continuum robots. The paper is easy to understand with a detailed kinematic model and simulation analysis. However, some concerns should be addressed before publishing.
1. The contribution of this work should be clearer. It’s recommended to give a table to compare with the existing technologies, such as dimension, accuracy or control.
2. The authors claim that the proposed robot can achieve better accuracy and control performance. Can you demonstrate some potential applications, such as localized laser ablation, to demonstrate the benefits of the robot's high precision and motion control?
3. Some related references are missing. Zhang, Tieshan, et al. "Millimeter‐scale soft continuum robots for large‐angle and high‐precision manipulation by hybrid actuation." Advanced Intelligent Systems 3.2 (2021): 2000189.
Minor editing of English language required.
